# HIF-1α is Overexpressed in Odontogenic Keratocyst Suggesting Activation of HIF-1α and NOTCH1 Signaling Pathways

**DOI:** 10.3390/cells8070731

**Published:** 2019-07-17

**Authors:** Natacha Malu Miranda da Costa, Caio Tadashi Saab Abe, Geovanni Pereira Mitre, Ricardo Alves Mesquita, Maria Sueli da Silva Kataoka, André Luis Ribeiro Ribeiro, Ruy Gastaldoni Jaeger, Sérgio de Melo Alves-Júnior, Andrew Mark Smith, João de Jesus Viana Pinheiro

**Affiliations:** 1Department of Buco-Maxillofacial Surgery and Traumatology and Periodontology, School of Dentistry of Ribeirão Preto, University of São Paulo, Av. do Café-Subsetor Oeste-11 (N-11), 14040-904 Ribeirão Preto, SP, Brazil; 2Cell Culture Laboratory, School of Dentistry, Federal University of Pará (UFPA), Cidade Universitária Prof. José da Silveira Neto-R. Augusto Corrêa, 1-Guamá, 66075-110 Belém, PA, Brazil; 3Department of Oral Surgery and Pathology, School of Dentistry, Universidade Federal de Minas Gerais, R. Prof. Moacir Gomes de Freitas, 688-Pampulha, 31270-901 Belo Horizonte, MG, Brazil; 4Department of Cell and Developmental Biology Institute of Biomedical Sciences, University of São Paulo, Av. Prof. Lineu Prestes, 1374-Butantã, 05508-900 São Paulo, SP, Brazil; 5Department of Microbial Diseases, Eastman Dental Institute, University College London, 256 Grays Inn Rd, London WC1X 8LD, UK

**Keywords:** odontogenic keratocyst, HIF-1α, hypoxia, oxygen

## Abstract

**Background: **The odontogenic keratocyst (OKC) is an odontogenic cyst that shows aggressive and intriguing biological behavior. It is suggested that a hypoxic environment occurs in OKC, which led us to investigate the immunoexpression and location of hypoxia-inducible factor 1-alpha (HIF-1α) and other hypoxia-related proteins. **Methods: **Twenty cases of OKC were evaluated for the expression of Notch homolog 1 (NOTCH1), HIF-1α, disintegrin and metalloproteinase domain-containing protein 12 (ADAM-12), and heparin-binding epidermal growth factor-like growth factor (HBEGF) by immunohistochemistry and compared to eight control cases of calcifying odontogenic cystic (COC), orthokeratinized odontogenic cyst (OOC), and normal oral mucosa (OM) in basal and parabasal layers. **Results: **In OKC, all the proteins tested were expressed significantly higher in both basal (except for NOTCH1 and HBEGF in OOC) and suprabasal epithelial layers compared to controls. Looking at the epithelial layers within OKC, we observed an increased NOTCH1 and HIF-1α expression in parabasal layers. **Conclusions: **These results suggest that hypoxia occurs more intensively in OKC compared to COC, OM, and OOC. Hypoxia appeared to be stronger in parabasal layers as observed by higher HIF-1α expression in upper cells. Overexpression of NOTCH1, ADAM-12, and HBEGF in OKC was observed, which suggests that microenvironmental hypoxia could potentially regulate the expression of hypoxia-related proteins, and consequently, its clinical and biological behavior.

## 1. Introduction

Odontogenic cysts and tumors arise from cells or tissues that are associated with tooth development and commonly located in jaw bones [1]. Among them, the odontogenic keratocyst (OKC), which was formerly known as keratocyst odontogenic tumor [2] and recently reclassified into the cyst category by the World Health Organization, is one of the most prevalent odontogenic lesions. The histology of OKC usually shows a thin parakeratinised epithelial lining with columnar cells in the basal layer which have focal reverse polarization [1]. Although it is now considered a developmental cyst, OKC presents an aggressive local behavior, high recurrence rates [3], and malignant transformation in very rare cases [4].

The distinguished behavior of OKC, which tends to grow in an anteroposterior direction causing limited bone expansion, suggests that complex molecular events can be underlying of its local invasiveness. Local invasiveness is a process that allows cystic or neoplastic cells to invade the surrounding tissues, which normally results from the degradation of the extracellular matrix (ECM) components. Thus, one of the first steps in tumor invasion is the breakdown of the basal membrane and degradation of the ECM by proteolysis [5]. It can be done by the formation of invadopodia, a finger-type structure formed by small membrane protrusions, which play a pivotal role during the early steps of local invasion.

Invadopodia are small invasive projections that are directly associated with localized proteolytic degradation of the ECM [6]. These microstructures are related to local invasiveness and tumor progression in both benign [7] and malignant [8] lesions. Local release of matrix metalloproteinases (MMPs), especially MMP-2 and MMP-9, are essential for tumor invasion [9]. MMP-2 [10], MMP-9 [11], and invadopodia-related proteins (e.g., cortactin, TKs5, TKs4 and MT1-MMP) have already been found to be overexpressed in OKC [7].

Recently, it was demonstrated that alterations in oxygen levels influence the biology of neoplastic cells [12]. Furthermore, it is already known that the concentration of oxygen varies in specific areas of the same tumor. Some areas show low oxygen concentrations that characterize tumor hypoxia, which has been associated with increased neoplastic invasion [12,13].

Under hypoxic conditions, a myriad of cellular responses is initiated, with a special interest to the Notch homolog 1 (NOTCH1) signaling. NOTCH1 signaling is a vital molecular pathway for several physiological [14] and pathological cellular processes, which is regulated by the cellular microenvironment [12]. Hypoxia leads to the initiation of NOTCH1 signaling pathway via activation of the hypoxia-inducible factor 1-alpha (HIF-1α) [15]. NOTCH1 signaling is directly related to an increased invasion of certain neoplasms [12,15].

HIF-1α is a transcription factor that responds to oxygen availability and considered a hypoxia marker [16]. In odontogenic lesions, HIF-1α overexpression has been associated with invasiveness [13] and cystic formation [17]. Furthermore, its activation can regulate certain genes linked to tumor growth and aggressiveness, such as the disintegrin and metalloproteinase domain-containing protein 12 (ADAM-12) [11,18].

ADAM-12 is a zinc-dependent metalloprotease [19] related to the pathogenesis of some neoplasms [20] and invadopodia formation under hypoxic conditions [12]. It is believed that ADAM-12 is capable of stimulating tumor growth by cleaving and releasing certain ligands that are biologically active, such as the heparin-bound epidermal growth factor (HBEGF) [12,21].

HBEGF is a powerful mitogen and also assists in the formation of invadopodia [22]. In addition, it can bind to cellular receptors associated with malignant transformation of a variety of human neoplasms [20,23].

In this study, we investigated the expression and location of NOTCH1, HIF-1α, ADAM-12, and HBEGF by immunohistochemistry in OKC. The expression of HIF-1α in different odontogenic cysts can provide additional evidence of whether and how hypoxia occurs in these lesions. The calcifying odontogenic cystic (COC), the orthokeratinized odontogenic cyst (OOC), and samples of the normal oral mucosa (OM) were used as controls to compare the relative expression of these proteins to OKC. 

## 2. Materials And Methods

### 2.1. Samples

Twenty cases of OKC were used in this study and compared to eight cases of COC, OOC, and OM each. The COC, OOC, and OM samples were used as controls. COC was included because it is an epithelial odontogenic cyst with a good prognosis and low relapse rates after simple enucleation [1]. OOC is a rare developmental odontogenic cyst that was previously considered as a variant of the OKC. Now, OOC is considered a unique entity that does not show a tendency to relapse due to the low local aggressiveness and is not associated with the nevoid basal cell carcinoma syndrome (NBCCS) [1]. The OM samples were used as representatives of a non-pathological epithelial oral tissue. These cases were obtained from the pathology records of the Department of Oral Pathology of the School of Dentistry, University Center of Pará (CESUPA, Belém-PA, Brazil). This study was conducted in accordance with the Declaration of Helsinki and approved by the Ethics Committee of the Human Research Ethics Committee of the Institute of Health Sciences in the Federal University of Pará (CAAE: 36572414.7.0000.0018, No. 877.322/2014).

### 2.2. Immunohistochemistry Assessment

Immunostaining was performed using an immunoperoxidase assay and the EnVision technique as previously described [13]. In this study, 5 μm histological sections obtained from samples and mounted on glass slides treated with 3-aminopropyltriethoxysilane (Sigma Chemical Corp, St. Louis, MO, USA) were used for immunostaining. Tissue sections were deparaffinised in xylol (Labsynth, São Paulo, BR) and hydrated in decreasing concentrations of ethanol (Labsynth^®^). Slides were then immersed in 20% H_2_O_2_ and methanol (Labsynth^®^) in a 1:1 ratio for 20 min to inhibit endogenous peroxidase activity. Subsequently, antigen recovery was performed in a citrate buffer (pH 6.0) in a Pascal pressure chamber (Dako, Carpinteria, CA, USA) for 30 s. Non-specific binding sites were blocked with 1% bovine serum albumin (BSA, Sigma^®^) in phosphate-buffered saline (PBS) for 1 h. Then, slides were incubated for 1 h with the following primary antibodies: anti-NOTCH1 (1:100, clone mN1A, Merck Millipore, Darmstadt, Germany), anti-HIF-1α (1:25, clone H1α67, Merck Millipore^®^), anti-ADAM-12 (1:150, catalog #bs-5847R, Bioss, Massachusetts, USA), and anti-HB-EGF (1:7.5, catalog #AF‑259‑NA, R&D Systems, Minnesota, EUA). Subsequently, sections were incubated for 30 min with EnVision Plus (Dako^®^) detection system. Diaminobenzidine (Dako^®^) was used as a chromogen. Sections were then counterstained with Mayer’s haematoxylin (Sigma^®^) and mounted with Permount (Fisher Scientific, Fair Lawn, NJ, USA). Primary antibody controls were done by Western blotting and secondary antibodies by replacing the primary antibodies by non-immune serum. Samples of the mucoepidermoid carcinoma were used as positive controls (not shown).

### 2.3. Length, Inflammation, and Immunostaining Assessment

Brightfield images from 10 arbitrary regions were selected from each sample and acquired using an AxioScope microscope equipped with an AxioCam HRC color CCD camera (Carl Zeiss, Oberkochen, Germany), using a 40× objective. The length of the basal layer was measured using the “Freehand line selections” tool from software ImageJ (public domain software developed by Wayne Rasband e NIMH, NIH, Bethesda, MD; http://rsbweb.nih.gov/ij/).

The inflammation score was assessed by counting the total number of inflammatory cells adjacent to the basal layer of the epithelium in each of the 10 images. Analysis of inflammation was carried out in grades: grade 0: no inflammation; grade 1: <15 cells/field; grade 2: 15–50 cells/field; and grade 3: >50 cells/field. The inflammatory score was calculated as the average of all fields examined. Samples of OKC, COC, and OM were divided into two groups according to the inflammation score: Group A: grades 0–2 (mild-to-moderate inflammation) and Group B: grade 3 (intense inflammation) [24].

Images were stacked to RGB and segmented using the H DAB vector of the color deconvolution plug-in of ImageJ. Channel 2 (color 2) was selected and images adjusted using the threshold tool. Areas stained with diaminobenzidine were identified and measured after selection using the freehand tool. The epithelial area was assessed in two different segments that corresponded to the basal and suprabasal layers. Results were expressed as the average percentage of labelling area (%) of the epithelial layers. Differences in the percentages of stained areas for the basal and suprabasal epithelial layers of OKC, COC, and OM were analysed.

### 2.4. Statistical Analysis

Data obtained from the experiments were analyzed using GraphPad Prism 5 software (GraphPad Software Inc., San Diego, CA, USA). Parametric ANOVA followed by Tukey’s post-test was used after analysis of normality to assess differences between the three groups: OKC, COC, and OM. Student’s *t*-test was used to compare basal and parabasal layers of protein expression within each lesion and between the two groups of inflammation scores. 

## 3. Results

### 3.1. Basal Layer Length and Degree of Inflammation

The basal layer length of the lesions varied between 5.9 mm and 15.7 mm in OKC (mean  =  8.8; SD =  1.8), 6.3 mm and 13.9 mm in COC (mean  =  8.9; SD  =  1.7), 5.2 mm and 17.4 in OOC, and 3.7 mm and 18.3 mm in OM (mean  =  8.9; SD  =  2.9). Inflammation assessment is detailed in Table 1. There was no difference among inflammation groups in OKC and the other lesions groups could not be analyzed due to the limited number of samples in the grade 3 group.

### 3.2. NOTCH1, HIF-1α, ADAM-12, and HBEGF were expressed in OKC, COC, OOC and OM

OKC, COC, OOC, and OM samples expressed all the studied proteins with different intensities (Figure 1, Figure 2, Figure 3 and Figure 4). NOTCH1 exhibited intense cytoplasmic immunostaining in OKC, showing increased expression in the upper layers of parabasal cells adjacent to the cystic lumen (Figure 1A,B). NOTCH1 nuclear expression was observed in some cells of the upper parabasal layers (Figure 1A). COC exhibited a weak and selective NOTCH1 expression in a few cells of the basal layer and a membrane staining in ghost cells (Figure 1C). OM samples showed a weak NOTCH1 expression, especially in the basal and suprabasal layers adjacent to the basal layer.

HIF-1α showed a strong immunolabelling in the upper layers of parabasal cells adjacent to the cystic lumen similar to what was observed with NOTCH1 in OKC (Figure 2A,B). Staining was nuclear and cytoplasmic and also observed in central epithelial islands of cystic formation (Figure 2C). A weak nuclear but not cytoplasmic staining was found in basal layer cells (Figure 2D). COC exhibited a very weak HIF-1α expression in a few parabasal cells and in selected cells of the basal layer (Figure 2E). In OM samples, HIF-1α expression appeared stronger than in COC but weaker than in OKC (Figure 2F). A very weak HIF-1α expression was found in OOC, similarly to what was observed in COC, but with a more evident expression in the basal layer (Figure 2G). 

ADAM-12 and HBEGF showed a similar pattern of expression in each group of lesions (Figure 3 and Figure 4). ADAM-12 and HBEGF were expressed in the cytoplasm of all epithelial strata of OKC (Figure 3A,B and Figure 4A,B). There was an even distribution in all the epithelial layers for both proteins in OKC. ADAM-12 expression in COC was generally weak, nuclear and cytoplasmic immunolocation was observed in the basal layer, while a less evident cytoplasmic expression was seen in the upper parabasal cells. However, HBEGF expression was predominantly cytoplasmic in COC, with no staining in ghost cells for both proteins. Expression of ADAM-12 and HBEGF in OM samples showed a comparable pattern seen in COC with similar weak staining. In OOC, ADAM-12 expression was very weak in all layers (Figure 3E), while HBEGF expression was moderate in the basal layer but very weak in parabasal layers (Figure 4E).

### 3.3. Expression of NOTCH1, HIF-1α, ADAM-12, and HBEGF is higher in OKC when compared to COC, OM, and OOC

OKC overexpresses NOTCH1, HIF-1α, ADAM-12, and HBEGF in either basal (except for NOTCH1 and HBEGF in OOC) and parabasal layers when compared to COC, OM, and OOC (Figure 5A,B). NOTCH1 and HIF-1α were more expressed in parabasal layers of OKC (Figure 5C). ADAM-12 and HBEGF did not show a statistical difference within their epithelial layers. No differences were observed between the basal and parabasal epithelial layers in COC (Figure 5D) and OOC (Figure 5F) for all proteins studied. However, OM samples showed differences only for the expression of ADAM-12, with predominant staining in the basal layer (*p* > 0.0001) when compared to the parabasal layers (Figure 5E).

## 4. Discussion

OKC has been investigated by several studies trying to elucidate the mechanisms associated with its intriguing biological and invasive behavior. Recently, the invadopodia-associated proteins (cortactin, MT1-MMP, Tks4, and Tks5) were described in this lesion [7], but what causes the invasive mechanism in this lesion is still unclear. In this study, the role of hypoxia as a microenvironmental factor influencing OKC biology was investigated as a possible cause. 

Our results provide evidence of increased expression of NOTCH1, HIF-1α, ADAM-12, and HBEGF in OKC when compared to COC, OM, and OOC. Interestingly, when comparing the basal and parabasal layers of OKC, we observed predominant staining of NOTCH1 and HIF-1α in the epithelial layers near to the cyst lumen. It suggests an increased hypoxic environment when the cells are distant from the oxygen supply provided by the surrounding connective tissue. It links tissue hypoxia, characterized here by higher expression of HIF-1α in parabasal layers than in the basal layer, to biological activity, the overexpression of NOTCH1. These findings support the evidence of tissue hypoxia in the upper suprabasal cells in OKC, wherein the expression pattern resembles the pattern also seen in NOTCH1. Although a cause–effect relationship cannot be determined by the methods used here (i.e., immunohistochemistry), HIF-1α overexpression is known to initiate the NOTCH1 signaling pathway [15], suggesting that a possible biological mechanism is triggered by tissue hypoxia.

OKC and OOC are histologically similar, showing a main difference in the type of keratin layer. In OKC, the keratin layer shows nucleated keratinocytes, while in the OOC, the keratin component is nuclei-free [1]. Prior to OOC being recognized as a different entity, OKC was classified as orthokeratinized or parakeratinized, and the last considered the most aggressive form. Here, the proteins studied and associated with more aggressive behavior were more expressed in OKC than in OOC, except for NOTCH1 and HBEGF in the basal layer that showed no differences. 

Activation of NOTCH1 results in its release from the plasma membrane and translocation to the nucleus [24], resulting in the activation of the transcription of several genes. These genes have been previously associated with tumor growth, by either increasing cell proliferation or reducing cell death [25]. Here, we observed nuclear NOTCH1 immunostaining in specific regions of the basal layer of OKC cells. This nuclear staining suggests NOTCH1 nuclear translocation and gene activation. It may result in the transcription of genes related to cell proliferation and invasion [15], possibly by inducing the activation of invadopodia-related genes [12]. A previous study has shown the overexpression of invadopodia-related proteins in the OKC basal layer [7]. Thus, these findings suggest that hypoxia occurs in OKC, which overexpresses NOTCH1. NOTCH1 is found in the nucleus of basal cells, suggesting its activation as a possible biological event influencing OKC invasion.

NOTCH1 signaling is known to promote the survival and proliferation of cancer cells, which is potentiated by hypoxia through stabilization of the active form of NOTCH1 by HIF-1α [12]. In this investigation, a higher expression of HIF-1α was identified in OKC when compared to COC, OM, and OOC. 

Studies have shown that the nuclear localization of HIF-1α is associated with the formation of invadopodia under hypoxic conditions, and subsequently, more aggressive behavior in some types of tumors [12,13]. It can be hypothesized that a hypoxic environment within the upper layers of OKC results in the activation of HIF-1α and NOTCH1. Together, they would induce the transcription of genes that will further lead to invadopodia formation.

We also observed a higher expression of NOTCH1 and HIF-1α in the parabasal layers in relation to the basal layer in OKC. In parabasal layers, both proteins showed intense nuclear and cytoplasmic immunostaining [24]. HIF-1α is classically known to be transient in the cytoplasm, which is quickly degraded by the proteasome. In this case, the cytoplasmic labelling of HIF-1α was probably a consequence of the proteins responsible for its degradation in cytoplasm having reached their saturation point, resulting in higher cytoplasmic levels [25].

Additionally, increased expression of NOTCH1 and HIF-1α in the parabasal layers may be connected to cellular mechanisms involved in cystogenesis, such as hypoxia and apoptosis [17]. Under hypoxic conditions, the *p53* tumor suppressor gene is activated via HIF-1α signaling [26]. It is known that *p53* is directly associated with apoptosis and cell proliferation [27], events that are fundamental for cystic formation. Furthermore, the regulation of NOTCH1 has been shown to occur via *p53* [27]. 

Here, the increased expression of NOTCH1 and HIF-1α observed in OKC can trigger the transcription of a myriad of host genes, including the metalloprotease *ADAM-12* [12]. The identification of high cytoplasmic levels of ADAM-12 in all epithelial layers in OKC supports this hypothesis. Inhibition of ADAM-12 has previously been shown to reduce HBEGF cleavage and cell migration, while its overexpression is linked to ADAM-12-mediated HBEGF cleavage and EGFR dysregulation [28]. The expression pattern for HBEGF was similar to that of ADAM-12. ADAM-12 probably induces the release of HBEGF in OKC. This could potentially lead to increased proliferation and proteolytic activity of OKC cells.

In addition, ADAM-12 has also been directly implicated in the formation of invadopodia through its cytoplasmic tail and activation of Src proteins kinases. Src proteins participate in the formation and activation of the invadopodia [29], which has the potential to contribute to the locally aggressive behavior of OKC.

In summary, our results suggest that hypoxia occurs more intensively in OKC than in COC, OM, and OOC. In OKC, hypoxia appearred to be more intense in parabasal layers near to the cystic lumen than in the basal layer, as observed by higher HIF-1α expression in these regions of the lesion. A similar pattern of expression within OKC epithelial layers was observed for NOTCH1, suggesting the activation of NOTCH1 by HIF-1α. ADAM-12 and HBEGF were found to be overexpressed in OKC when compared to COC, OM, and OOC, a possible result of downstream activation of the NOTCH1 and HIF-1α signaling pathways and transcription. Together, these results support a biological role for microenvironmental hypoxia in OKC that may contribute to its intriguing biological behavior. Although the HIF-1α expression pattern seen in OKC can be suggestive of tissue hypoxia, the method used in this study cannot determine that the altered expression of the associated proteins studied here were regulated by it. Further mechanistic studies may provide supportive evidence of the role of hypoxia in OKC clinical behavior.

## Figures and Tables

**Figure 1 cells-08-00731-f001:**
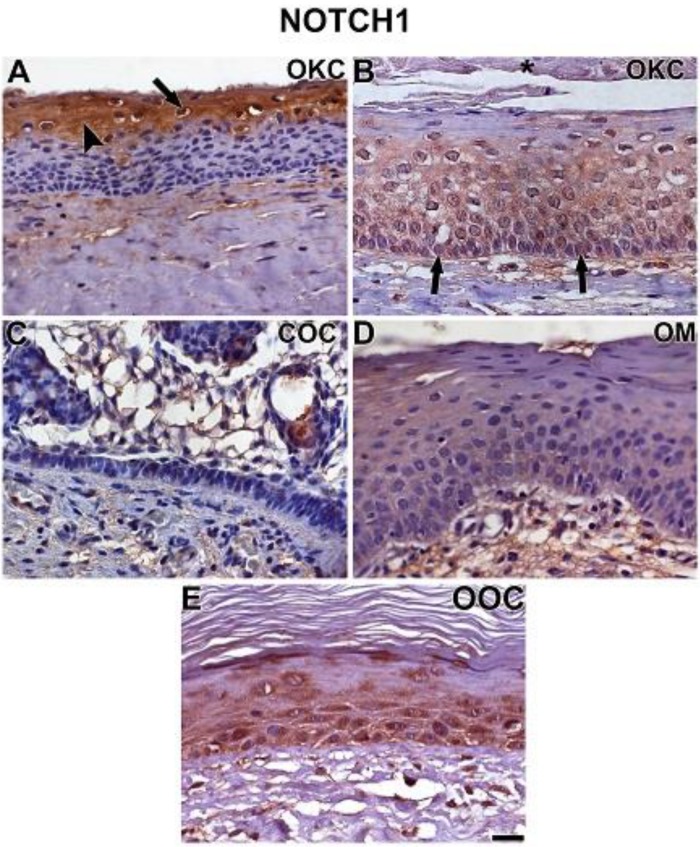
Immunostaining of NOTCH1 in OKC (**A**,**B**), COC (**C**), OM (**D**), and OOC (**E**). NOTCH1 exhibited intense and uniform cytoplasmic (**A**, arrowhead) staining in epithelial layers adjacent to the cystic lumen. Strong nuclear staining was observed in some cells of the upper parabasal layers (**A**, arrow). In some cases, basal (**B**, arrows) and parabasal layers showed NOTCH1 expression. COC exhibited a weak and selective NOTCH1 expression in a few cells of the basal layer. Staining of ghost cells was observed around the plasma membrane (**C**). A very weak NOTCH1 expression was seen in OM (**D**), especially in the basal and lower suprabasal layers (**D**). OOC showed basal and suprabasal staining (**E**). Scale bar: 20 µm.

**Figure 2 cells-08-00731-f002:**
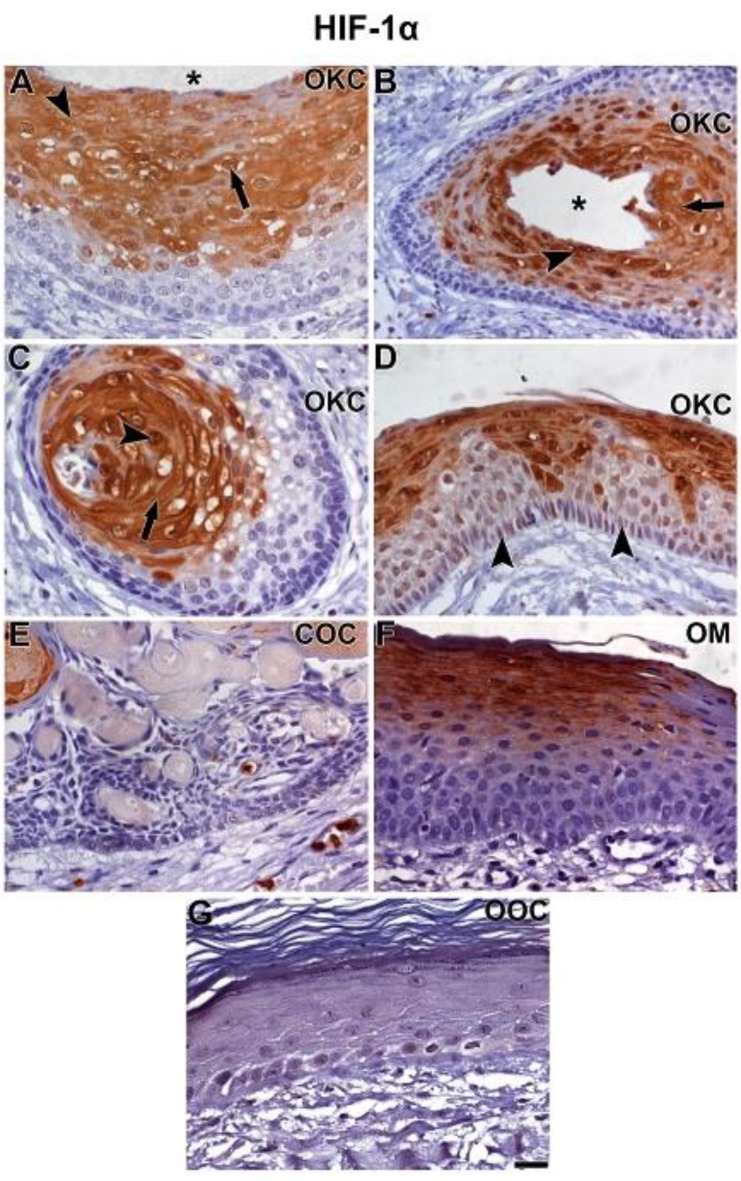
Immunolabelling of HIF-1α in OKC (**A**–**D**), COC (**E**), OM (**F**), and OOC (**G**). HIF-1α immunoexpression showed strong nuclear (**A** and **B**, arrow) and cytoplasmic (**A** and **B**, arrowhead) staining in OKC, mainly in the epithelial layers near to the cyst lumen (**A**, asterisk) and in areas of cyst formation (**B**, asterisk). In the center of epithelial islands formed by the odontogenic epithelium, the so-called pupal cysts, intense cytoplasmic (**C**, arrow) and nuclear (**C**, arrowhead) staining was seen. Weak nuclear but not cytoplasmic staining was found in basal layer cells (**D**, arrowheads). COC exhibited a very weak HIF-1α expression in a few parabasal cells. Staining in the basal layer was very weak and in a few selected cells. (**E**) HIF-1α expression in OM appeared to be stronger than in COC but still weaker than in OKC. The pattern of HIF-1α expression was more similar to that seen in OKC, where staining intensity increased towards upper cells of the parabasal layers. HIF-1α expression in OOC was very weak, observed in selected basal and suprabasal cells (**G**). Scale bar: 20 µm.

**Figure 3 cells-08-00731-f003:**
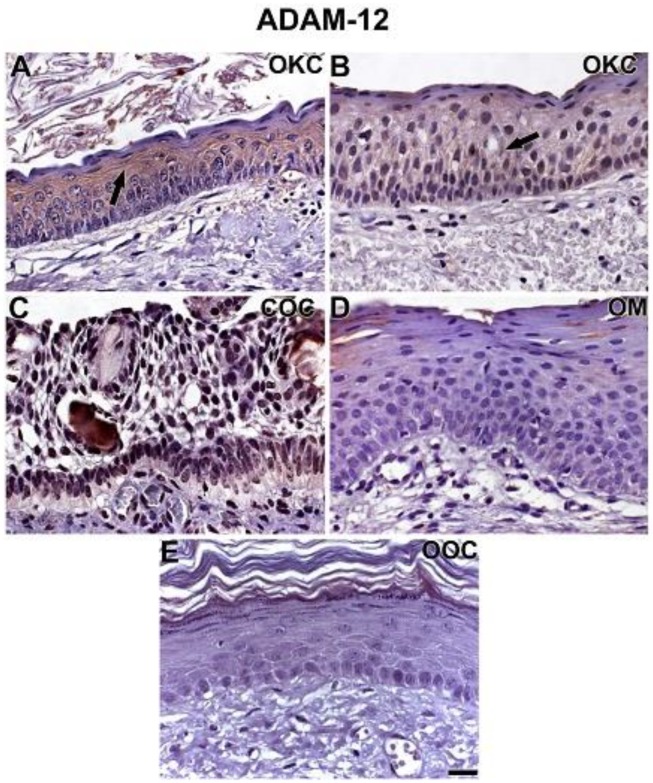
Immunolabelling of ADAM-12 in OKC (**A**,**B**), COC (**C**), OM (**D**), and OOD (**E**). ADAM-12 was expressed as dots in the cytoplasm of all epithelial strata (**A**,**B**, arrows) in OKC. There was an even distribution in all the epithelial layers. ADAM-12 expression in COC was weak, as seen in the cytoplasm of basal cells, and less evident cytoplasmic expression was found in parabasal cells. Staining of ghost cells was not observed. (**C**) ADAM-12 expression in OM samples (**D**) and in OOC (**E**) showed a similar pattern seen in COC, the overall expression appeared to be very weak in both. Scale bar: 20 µm.

**Figure 4 cells-08-00731-f004:**
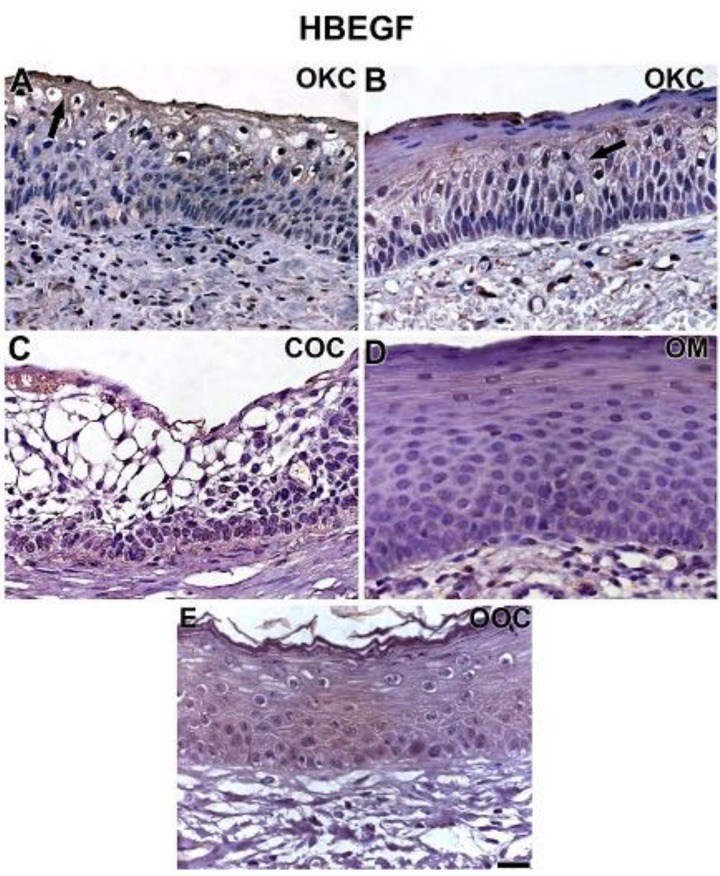
Immunolabelling of HBEGF in OKC (**A**,**B**), COC (**C**), OM (**D**), and OOC (**E**). Generally speaking, HBEGF expression resembled the pattern observed with ADAM-12 for all groups. HBEGF was expressed in the cytoplasm of all epithelial strata (**A**,**B**), especially in the periphery of the cytoplasm (**A**,**B**, arrows). There was an even distribution in all the epithelial layers. HBEGF expression in COC was predominantly cytoplasmic, with no expression in ghost cells (**C**). In OM samples, immunostaining was seen in the cytoplasm of the basal and predominantly in lower cells of the parabasal layers (**D**). HBEGF was predominantly found in the cytoplasm of basal cells in OOC; however, its expression was weak in parabasal layers (**E**). Scale bar: 20 µm.

**Figure 5 cells-08-00731-f005:**
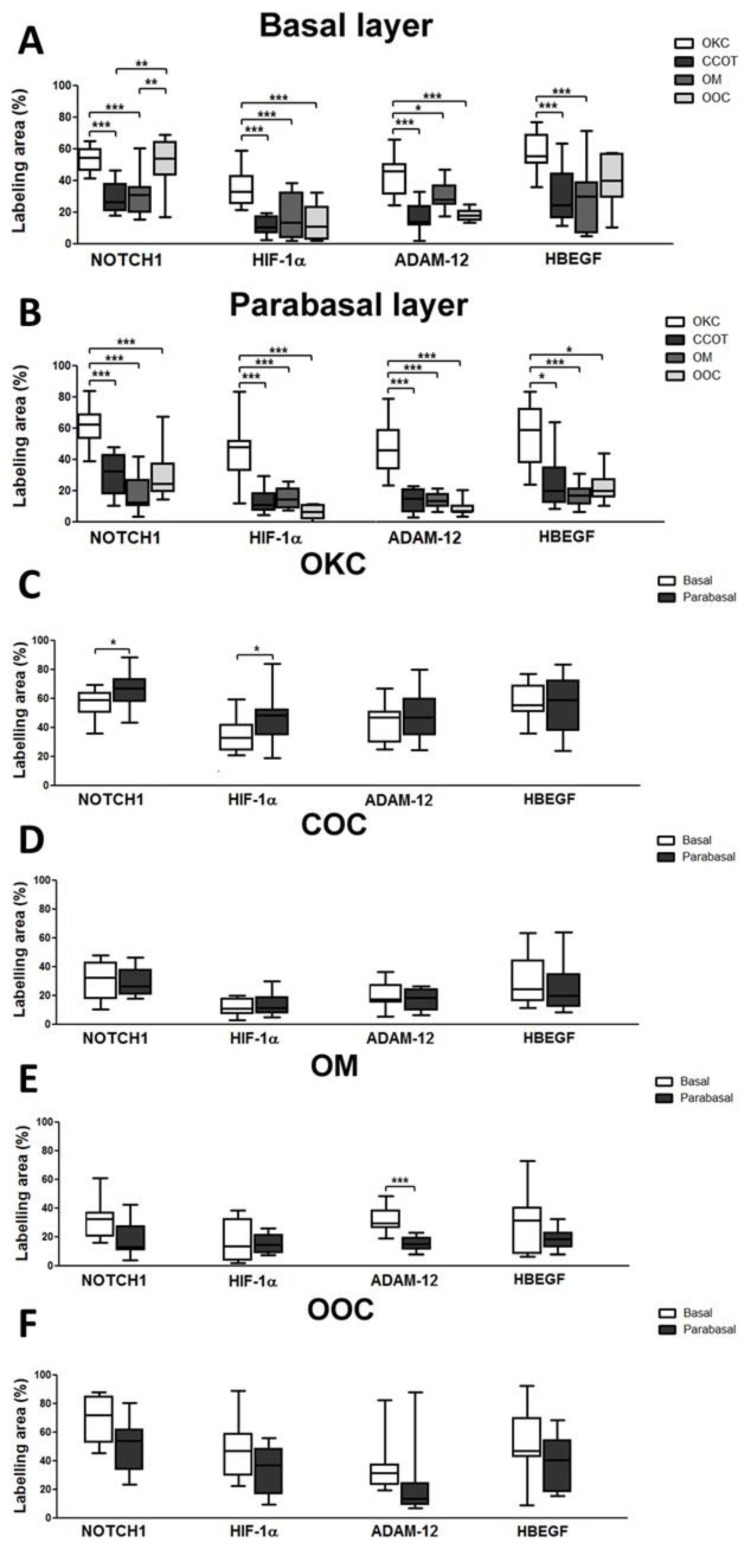
Comparison of NOTCH1, HIF-1α, ADAM-12, and HBEGF immunolabelling in basal (**A**) and parabasal (**B**) layers between OKC, COC, OM, and OOC. OKC overexpressed all the studied protein when compared to COC, OM, and OOC in either basal (**A**, except for NOTCH1 and HBEGF in OOC) and parabasal (**B**) layers. Immunoexpression between basal and parabasal layers within the same lesions was also evaluated in OKC (**C**), COC (**D**), OM (**E**), and OOC (**F**). Differences were only seen for NOTCH1 and HIF-1α in OKC and ADAM-12 in OM. In OKC, NOTCH1, and HIF-1α were more expressed in parabasal layers, while ADAM-12 in OM showed a higher expression in the basal layer. * *p* < 0.05; ** *p* < 0.01; *** *p* < 0.001.

**Table 1 cells-08-00731-t001:** Length of the basement membrane and inflammation grade in OKC, COC and OM samples.

	Length of the Basement Membrane (mm)				Inflammation Grade				
	Mean	Min	Max	SD	0	1	2	3	A × B
OKC	8.8	5.9	15.7	1.8	-	4	12	4	ns
COC	8.9	6.3	13.9	1.7	-	-	7	1	na
OOC	8.4	5.2	17.4	2.1	-	5	1	2	na
OM	8.9	3.7	18.3	2.9	-	6	2	-	na

Abbreviation: OKC, odontogenic keratocystic; COC, calcifying odontogenic cystic; OOC, orthokeratinized odontogenic cyst; OM, oral mucosa; SD, standard deviation; A × B describes the summary of statistical analysis for NOTCH, HIF-1α, ADAM-12, HBEGF between A, inflammation assessment Group A—grades 0–2 (mild-to-moderate) and B, inflammation assessment group B—grade 3 (intense); ns, non-significant; na, not applicable.

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
