# Peer review of "HIF-1α is Overexpressed in Odontogenic Keratocyst Suggesting Activation of HIF-1α and NOTCH1 Signaling Pathways"

_cells, 2019, doi:10.3390/cells8070731_

Round 1

Reviewer 1 Report

you should have groups with similar number of cases

Author Response

English language and style

(x) Extensive editing of English language and style required 
( ) Moderate English changes required 
( ) English language and style are fine/minor spell check required 
( ) I don't feel qualified to judge about the English language and style

Reply: We have carried out an extensive language and style revision.

Yes

Can be improved

Must be improved

Not applicable

Does the introduction provide   sufficient background and include all relevant references?

(x)

( )

( )

( )

Is the research design appropriate?

( )

(x)

( )

( )

Are the methods adequately described?

( )

(x)

( )

( )

Are the results clearly presented?

( )

(x)

( )

( )

Are the conclusions supported by the   results?

(x)

( )

( )

( )

Reply: We have carried out an extensive revision of language and contents.

Comments and Suggestions for Authors

you should have groups with similar number of cases

Reply: It would be an ideal condition fi we could have all groups with the same number of cases, however, orthokeratinized odontogenic cyst and calcifying odontogenic cyst are not common lesions and we used all we had access. The focus is on the odontogenic keratocyst, that is why we used a higher number of samples, which resulted in a convenience sampling for conducting this research.

Reviewer 2 Report

The article is interesting, but fails to compare the study groups. The comparison between keratocyst with oral mucosa and with COC is interesting but not the most adequate.

It would be much more interesting to compare a KOC group with parakeratinization and orthokeratinization. In fact, this aspect can be recovered and introduced into the study.

Another option would have been to do this type of analysis in KOC prior to decompression and after a period of three months of decompression.

In both situations (the type of keratinization, as well as decompression) there is evidence that it improves the clinical behavior of the lesion, so confirming that both situations are related to a lower expression of "hypoxic" proteins could be interesting.

The differences between the oral mucosa and the COC with the KOC are very wide and to base that the clinical behavior only depends on the high expression of these proteins is something risky.

Therefore, the authors must reform the article and include among the variables studied the keratinization type of the KOC. In addition, they should include decompression within the discussion (along with the type of keratinization) and their influence on clinical behavior. Finally, they should discuss the limitations of the study in relation to the chosen groups.

Author Response

English language and style

( ) Extensive editing of English language and style required 
( ) Moderate English changes required 
(x) English language and style are fine/minor spell check required 
( ) I don't feel qualified to judge about the English language and style 

Reply: We have carried out an extensive language and style revision.

Yes

Can be improved

Must be improved

Not applicable

Does the introduction provide sufficient   background and include all relevant references?

( )

(x)

( )

( )

Is the research design appropriate?

( )

( )

(x)

( )

Are the methods adequately described?

( )

( )

(x)

( )

Are the results clearly presented?

( )

( )

(x)

( )

Are the conclusions supported by the results?

( )

( )

(x)

( )

Reply: We have carried out an extensive revision of language and contents.

Comments and Suggestions for Authors

The article is interesting, but fails to compare the study groups. The comparison between keratocyst with oral mucosa and with COC is interesting but not the most adequate.

It would be much more interesting to compare a KOC group with parakeratinization and orthokeratinization. In fact, this aspect can be recovered and introduced into the study.

Reply: We thank to the reviewer for this suggestion. An extra group was added (orthokeratinized odontogenic cyst) and the entire manuscript was revised for this new content.

Another option would have been to do this type of analysis in KOC prior to decompression and after a period of three months of decompression.

In both situations (the type of keratinization, as well as decompression) there is evidence that it improves the clinical behavior of the lesion, so confirming that both situations are related to a lower expression of "hypoxic" proteins could be interesting.

Reply: The response to treatment is something very interesting that the literature still requires more investigation. However, comparing the protein expression prior and after decompression focusses on treatment rather than on the clinical behaviour, the last the main goal of this paper. As said before, we added a group for comparing the types of keratinization (orthokeratinized odontogenic cyst).

The differences between the oral mucosa and the COC with the KOC are very wide and to base that the clinical behavior only depends on the high expression of these proteins is something risky.

Therefore, the authors must reform the article and include among the variables studied the keratinization type of the KOC. In addition, they should include decompression within the discussion (along with the type of keratinization) and their influence on clinical behavior. Finally, they should discuss the limitations of the study in relation to the chosen groups.

Reply: In this new version, we revised some statements and discussed the keratinization type of the cysts and the limitations of the study.

Reviewer 3 Report

The present work is simple and well written, clear and brief. It is a methodologically simple work that is only based on immunohistochemical findings, however it yields relevant information that it serves to propose hypotheses. There are points that should be clarified by the authors:

1.The authors must specify the antibody clone used.

2.The immunohistochemistry technique should supported on positive and negative controls, did the authors use them?

3.In Figure 1B the reviewer does not clearly observe the nuclear positivity, it is recommended to place it another photo where nuclear immunostaining is evident, or take the result as negative.

4.In figure 3B the membrane positivity is not clear. Generally the membrane positivity in epithelial cells should be more evident resembling a network. Either a strong membrane staining should shown or the result must be considered negative membrane. The same applies in figure 4.

5.This work is only observational, and no hypothesis is tested. In order to assert something, the authors must have another methodological approach that should be predominantly experimental, based not only on immunohistochemical findings. The authors can only suggest and speculate possible hypotheses to be tested with another experimental methodology. That limitation of the study must be clear. The authors must specify what the limitations of the study were.

Author Response

English language and style

( ) Extensive editing of English language and style required 
( ) Moderate English changes required 
( ) English language and style are fine/minor spell check required 
(x) I don't feel qualified to judge about the English language and style 

Reply: We have carried out an extensive language and style revision.

Yes

Can be improved

Must be improved

Not applicable

Does the introduction provide sufficient   background and include all relevant references?

(x)

( )

( )

( )

Is the research design appropriate?

(x)

( )

( )

( )

Are the methods adequately described?

(x)

( )

( )

( )

Are the results clearly presented?

( )

( )

(x)

( )

Are the conclusions supported by the results?

( )

(x)

( )

( )

Reply: We have carried out an extensive revision of language and contents.

Comments and Suggestions for Authors

The present work is simple and well written, clear and brief. It is a methodologically simple work that is only based on immunohistochemical findings, however it yields relevant information that it serves to propose hypotheses. There are points that should be clarified by the authors:

1.The authors must specify the antibody clone used.

Reply: We have added the clone or catalogue number of all antibodies.

2.The immunohistochemistry technique should supported on positive and negative controls, did the authors use them?

Reply: We have used controls for primary and secondary antibodies. The primary antibodies controls method was done by western blotting and the secondary antibody by removing the primary antibodies of samples. The positive control was the mucoepidermoid carcinoma. These figures were not shown.

3.In Figure 1B the reviewer does not clearly observe the nuclear positivity, it is recommended to place it another photo where nuclear immunostaining is evident, or take the result as negative.

Reply: We have changed it and now it is described only in Figure 1A.

4.In figure 3B the membrane positivity is not clear. Generally the membrane positivity in epithelial cells should be more evident resembling a network. Either a strong membrane staining should shown or the result must be considered negative membrane. The same applies in figure 4.

Reply: We have revised this statement and we agreed with the reviewer. This information has been removed.

5.This work is only observational, and no hypothesis is tested. In order to assert something, the authors must have another methodological approach that should be predominantly experimental, based not only on immunohistochemical findings. The authors can only suggest and speculate possible hypotheses to be tested with another experimental methodology. That limitation of the study must be clear. The authors must specify what the limitations of the study were.

Reply: In this new version, we revised some statements and added the limitations of this study.

Round 2

Reviewer 2 Report

The article has improved a lot. I believe that it can be published in its current form. Congratulations.